# Vitamin D Status in Patients before Thyroidectomy

**DOI:** 10.3390/ijms24043228

**Published:** 2023-02-06

**Authors:** Dominika Maciejewska-Markiewicz, Joanna Kochman, Karolina Jakubczyk, Piotr Bargiel, Zbigniew Szlosser, Ewa Stachowska, Marta Markowska, Anna Bucka, Norbert Czapla, Jan Petriczko, Anna Surówka, Sonia Hertman, Piotr Puchalski, Piotr Prowans

**Affiliations:** 1Department of Human Nutrition and Metabolomics, Pomeranian Medical University, 71-460 Szczecin, Poland; 2Department of Plastic, Endocrine and General Surgery, Pomeranian Medical University, 72-010 Szczecin, Poland; 3Department of Pediatrics and Infectious Diseases, Region Hospital in Szczecin, 71-460 Szczecin, Poland

**Keywords:** vitamin D, thyroidectomy, 25-hydroxycholecalciferol, 1,25-dihydroxycholecalciferol

## Abstract

Thyroid neoplasms (tumors) are the most common pathology of the endocrine system that requires surgery, and in most cases changes are benign. The surgical treatment of thyroid neoplasms consists in total, subtotal, or one lobe excision. Our study aimed to assess the concentration of vitamin D and its metabolites in patients before thyroidectomy. The study included 167 patients with thyroid pathology. Before the thyroidectomy procedure calcidiol (25-OHD), calcitriol (1,25-(OH)_2_D), and vitamin D binding protein (VDBP), as well as basic biochemical parameters, were measured using an enzyme-linked immunosorbent assay kit. Data analysis showed that the cohort of patients has a significant 25-OHD deficiency and proper concentration of 1,25-(OH)_2_D. Before the surgery, more than 80% of patients have extreme vitamin D deficiency (<10 ng/mL), and only 4% of the study group has proper 25-OHD concentration. Patients undergoing thyroidectomy are exposed to many complications, including calcium reduction. Our research has shown that patients prior to surgery have a marked vitamin D deficiency, an indicator that may affect their subsequent convalescence and prognosis. The results suggest that determination of vitamin D levels prior to thyroidectomy may be useful for potential consideration of supplementation when vitamin D deficiency is marked and needs to be incorporated into the good clinical management of these patients.

## 1. Introduction

Thyroid diseases are a common health problem worldwide. Recent evidence has demonstrated an association between low vitamin D status and Hashimoto’s thyroiditis, Graves’ disease, and thyroid neoplasms [1]. Thyroid neoplasms (tumors) are the most common pathology of the endocrine system that requires surgery, and in most cases these changes are benign. Thyroid nodules are diagnosed in 1% of men and 5% of women worldwide [2]. Nodule incidence increases with age, iodine deficiency, and radiation exposure, and they are more frequent in women. Numerous studies suggest that the prevalence of thyroid tumors is diagnosed in 8 to 65% of autopsy data, 19 to 35% in ultrasound, and 2 to 6% in palpation examination [3]. Surgical treatment of benign nodules is considered mainly in the case of non-toxic goiter, especially nodular goiter [4,5]. The prevalence of thyroid cancer is estimated at 1–2% of thyroid neoplasms The surgical treatment of thyroid tumors consists of total, subtotal, or one lobe excision. In some cases, it is necessary to extend the scope of surgery to adjacent lymph nodes. Based on preoperative aspiration biopsy, it is possible to remove one lobe of the thyroid gland in benign or low-risk neoplasms [6,7]. The presence of multifocal lesions required the entire thyroid gland removal. The basics of thyroid cancer treatment are associated with the removal of the thyroid gland and excision of the middle neck lymph nodes. In the case of a microcarcinoma, the removal of one affected lobe of the thyroid gland is recommended [8,9]. Thyroidectomy has significant consequences and possible complications. The thyroid gland is located near the retrograde laryngeal nerves and the parathyroid glands. The retrograde laryngeal nerves are responsible for the mobility of the vocal cords. The parathyroid glands are responsible for the metabolism of calcium in the body. Patients after total thyroidectomy and most of them after lobectomy require the use of thyroid hormones for the rest of their lives. In some cases, surgical treatment is complicated by hypoparathyroidism and retrograde laryngeal paralysis [10,11].

Postoperative hypoparathyroidism is associated with a significantly reduced quality of life [12]. It is estimated that 7 to 30% of patients undergoing total thyroidectomy will have at least temporary hypocalcemia; therefore, it is important to maintain a consistent protocol for calcium management after total or subtotal thyroidectomy to minimize calcium reduction [13]. Permanent decreased calcium levels following a parathyroidectomy may contribute to a number of conditions related to renal failure and an increased risk of death [14,15]. One of the most important factors maintaining the adequate status of calcium in the blood is vitamin D. In the general population, the prevalence of vitamin D deficiency ranges from 20 to 80% [16]. Vitamin D level had a significant effect on hypocalcemia after thyroidectomy. Although post-thyroidectomy hypocalcemia is multifactorial, severe vitamin D deficiency is significantly associated with the development of biochemical and clinical hypocalcemia and its complication [1]. Choi et al. revealed that preoperative vitamin D deficiency is associated with a higher risk of hypocalcemia following total thyroidectomy, and the average concentration of vitamin D before surgery was 14.6 mg/mL [17]. Vitamin D supplementation can prevent further complications and reduce symptoms of transient hypocalcemia [1]. Therefore, our study aimed to assess the concentration of vitamin D in patients before thyroidectomy procedure.

## 2. Results

### 2.1. Vitamin D Status among All Patients before Thyroidectomy Procedure

Before the surgery, more than 80% of patients have extreme vitamin D deficiency (<10 ng/mL), and only 4% of the study cohort has proper 25-OHD concentration (Figure 1).

### 2.2. Vitamin D Status between Patients with Normal and Increased BMI

The analysis revealed that patients with higher BMI have higher concentrations of 1.25-(OH)_2_D and PTH. There were no significant changes in 25-OHD and VDBR concentration (Table 1).

### 2.3. Vitamin D Concentration and Type of Thyroid Tumor

There were no significant differences between the type of tumor/surgery and concentration of 25-OHD, 1,25-(OH)_2_D, and VDBP (Table 2).

### 2.4. Calcium Status after Surgery

Calcium concentration after surgery was significantly reduced (Table 3) and decreased below the normal range. The analysis revealed also a very weak, positive correlation between 25-OHD before surgery and calcium concentration after thyroidectomy (*p* < 0.04, RHO = 0.2).

## 3. Materials and Methods

### 3.1. Patients

The study included 167 patients with thyroid pathology who were patients of the Department of Plastic, Endocrine and General Surgery in Szczecin, between 2020 and 2021. The patients underwent total or partial thyroidectomy. The main reasons for the surgery were single thyroid nodules, nodular goiter, hyperthyroidism due to Graves Basedow’s disease, autonomic nodules, suspected cancer, and the diagnosis of thyroid cancer. The concentration of basic biochemical parameters was measured before a surgery to assess the hormones changes and calcium status. In the order to protect the retrograde laryngeal nerves, all procedures were performed with neuromonitoring. Postoperatively, each patient was administered L-thyroxine hormones appropriately matched to body weight, parathyroid hormone (PTH), and calcium level. Exclusion criteria were renal diseases, liver diseases, and intestinal diseases. The study protocol was approved by the ethics committee of the Pomeranian Medical University and conformed to the ethical guidelines of the 1975 Declaration of Helsinki (KB-0012/195/19). The volunteers provided written informed consent before the study. Patient characteristics are presented in Table 4.

### 3.2. Biochemical Analysis

All biochemical parameters were performed in the Laboratory of Independent Public Regional Hospital in Szczecin during routine analysis before thyroidectomy. For the determination of free calcium ion (ionized calcium), ion-selective electrodes was used. Calcidiol (25-OHD), calcitriol (1,25-(OH)_2_D), and vitamin D binding protein (VDBP) were measured using an enzyme-linked immunosorbent assay kit (EIAab science inc, Wuhan 430075, China).

### 3.3. Statistical Analysis

The statistical analysis was performed using the “R 4.0.3” software. The normality of continuous variables distribution by using Shapiro–Wilk test was evaluated, and non-parametric tests were used. The Mann–Whitney U test was used to analyze the differences between the groups. Data are presented as medians and interquartile ranges (IQR). The values of *p* < 0.05 were considered as statistically significant.

### 3.4. Vitamin D Suplemmentation

Patients were asked about cholecacyferol supplementation. A total of 15% of patients were supplementing vitamin D, with a total dose not exceeding 4000 IU/day, which is the highest acceptable dose for the general adult population.

## 4. Discussion

Vitamin D plays a crucial role in calcium homeostasis and the maintenance of bone health. First, Vitamin D binds to VDBP and transports to the liver. Then, hydroxylases convert inactive vitamin D to 25-OHD. In the next step 25-OHD is attached to VDBR and goes to the kidney, where an active metabolite (1,25-(OH)_2_D) is synthesized [18]. In contrast to 25-OHD, 1,25-(OH)_2_D has a short half-life period and is tightly regulated over a narrow range by PTH, calcium, and phosphate. The concentration of 1,25-(OH)_2_D is not a good predictor of vitamin D status unless the deficiency is severe. Therefore vitamin D status is evaluated according to 25-OHD concentration [19,20]. Our study revealed that almost all patients (96%) had insufficient 25-OHD concentration before the surgery procedure, and only 15% supplement vitamin D. It is estimated that patients with a lower concentration of 25-OH have a higher probability of hypocalcemia after thyroidectomy [21]. Tolone et al., revealed that compared to the control group, preoperative supplementation of vitamin D and calcium decreases the incidence of transient hypocalcemia after total thyroidectomy from 25.9% to 6.8% [22]. Calcium homeostasis depends on PTH and 1,25-(OH)_2_D concentration. PTH is responsible for the proper conversion of vitamin D and synthesis of active 1,25-(OH)_2_D. Lower PTH concentrations can reduce vitamin D conversion and decrease calcium absorption from the intestine [23]. Therefore, increased vitamin D before surgery can be important factor to maintain proper calcium concentration after surgery. Moreover, our study showed that patients with increased BMI has higher concentrations of 1,25-(OH)_2_D. This observation can be a consequence of the significant increase in PTH concentration. A high level of PTH is observed in obese and overweight patients. The correlation of PTH with a percentage of body fat seems to be independent of the plasma 25-OHD concentration, and the association between high fat level and PTH is still unclear [24]. However, one study suggests that increased PTH concentration in overweight and obese patients is associated with a lower level of 25-OHD and the need to intensify the conversion of 25-OHD to a biologically active form [25]. The PTH-vitamin D metabolism axis in our patients is presented in Figure 2

In recent decades, many experimental studies have shown a wide range of anticancer effects of vitamin D compounds [26]. Cell culture studies revealed decreased thyroid cancer growth after the administration of 1,25-(OH)_2_D or its analogs in several thyroid cancer cell lines. The antitumor effect of 1,25-(OH)_2_D on thyroid cancers can be associated with proliferation inhibition [27]. It has been shown that 1,25-(OH)_2_D inhibits the expression of *c-MYC*, the proto-oncogene, causing an increased percentage of cells in the G0-G1 phase [28]. Roskies et al., showed that patients undergoing thyroidectomy had a higher malignancy rates in the vitamin D deficient group, suggesting that a lower level of 25-OHD is a potentially modifiable risk factor for thyroid cancer [29]. Sahin et al., reported that patients with papillary thyroid carcinoma had significantly lower vitamin D levels than controls, and decreases in 25-OHD concentration were more prevalent in the study group [30]. Kim et al., investigated 548 female individuals who underwent a total thyroidectomy due to thyroid cancer; the preoperative 25-OHD concentration was significantly lower in patients with a tumor size of less than 1 cm [31]. Stepien et al., revealed a significantly lower concentration of 1,25-OH_2_D in thyroid cancer compared to healthy controls [32]. Penna-Martinez et al., revealed that patients with thyroid carcinoma had lower serum 1,25-(OH)_2_D levels compared to healthy controls; there were no significant differences in 25-OHD levels in these group of patients [33,34]. Our study group revealed no significant differences in any of vitamin D metabolites or VDBP between benign nodules and thyroid cancers. However, the concentration of 25-OH in both groups was extremely low, and all patients suffered from deep vitamin D deficiency. The concentration of 25-OHD in more than 80% of patients was less than 10 ng/mL and only 4% had the proper level of vitamin D. Low vitamin D levels before surgery can have negative effects on calcium concentration after the procedure. Our study revealed weak positive correlation (RHO = 0.2, *p* = 0.04) between 25-OHD before thyroidectomy and calcium concentration after surgery. Moreover, after the procedure calcium concentration was significantly lower and decreased below the normal range. Decreased concentrations of vitamin D are associated with several health outcomes, such as increased bone loss, risk of fracture, muscle weakness, or insulin resistance [35]. Mayank et al. showed that a pre-operative concentration of 25-OHD has a positive correlation with calcium concentration in the early period after thyroidectomy. Patients with a 25-OHD concentration less of 20 ng/mL are highly likely to develop hypocalcemia [36]. As long as we investigate how to exploit vitamin D in cancer prevention and treatment, the basic recommendation to aim at a sufficient vitamin D level is still valid [37].

## 5. Conclusions

Patients undergoing thyroidectomy are exposed to many complications, including abnormal calcium concentration. Adequate vitamin D levels before the surgery appears to be an important factor in maintaining and preventing proper calcium concentration. Our research has shown that patients prior to surgery have a marked vitamin D deficiency, an indicator that may affect their subsequent convalescence and prognosis. Only 15% of them supplement vitamin D before surgery. The results suggest that the determination of vitamin D levels prior to thyroidectomy may be useful for potential consideration of supplementation when vitamin D deficiency is marked and needs to be incorporated into the good clinical management of these patients.

## 6. Limitations

Our study has several limitations. First of all, we did not have access to long-term measurements of calcium or 25-OHD concentration. Second of all, the size of the study group was determined by the number of patients admitted to the hospital. We did not carry out predetermined sample size estimates considering underlying hypotheses and sufficient underlying statistical power to answer them. Moreover, we do not have data about long-term complications after thyroidectomy procedure.

## Figures and Tables

**Figure 1 ijms-24-03228-f001:**
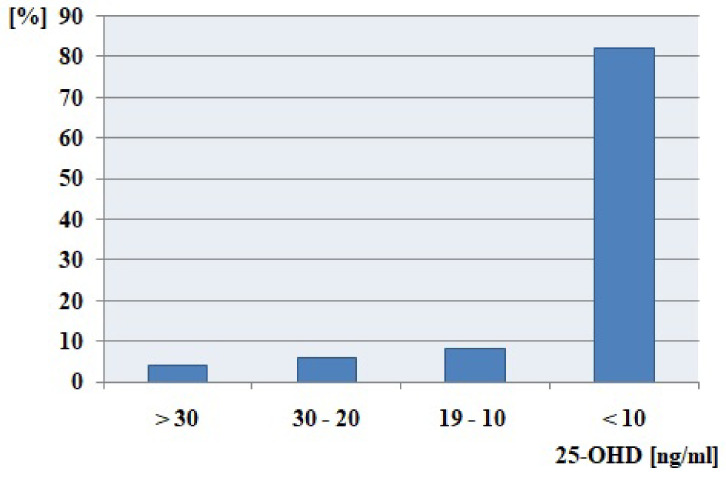
25-OHD deficiency level among all patients before thyroidectomy procedure.

**Figure 2 ijms-24-03228-f002:**
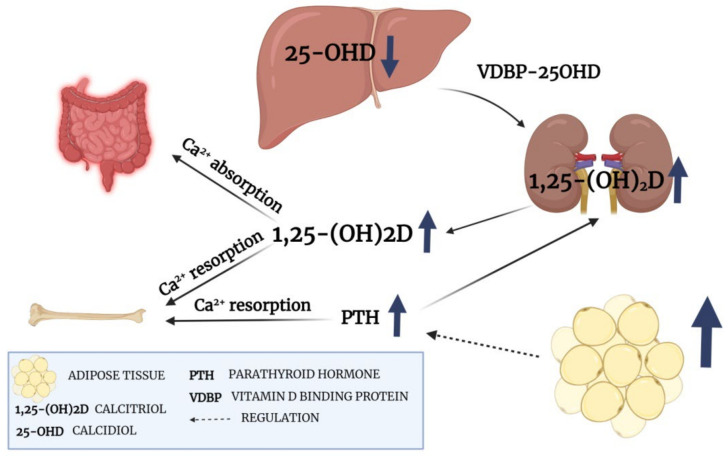
PTH-vitamin D metabolism axis (created with BioRender).

**Table 1 ijms-24-03228-t001:** Vitamin D status between patients with normal and increased BMI.

Vitamin D Status	Normal BMI		Incresed BMI		*p*
	Median	IQR	Median	IQR	
25-OHD [ng/mL]	8.55	40.04	4.25	44.21	0.17
1.25-(OH)_2_D [pg/mL]	44.77	5.73	53.4	8.03	0.01
VDBP [ng/mL]	29.77	22.11	30.37	23.20	0.57
PTH [pg/mL]	15.45	17.72	24.4	24.6	0.02

**Table 2 ijms-24-03228-t002:** Vitamin D status between patients with benign nodules and thyroid cancer.

Vitamin D Status	Benign Nodules		Thyroid Cancers		*p*
	Median	IQR	Median	IQR	
25-OHD [ng/mL]	4.96	2.51	4.25	2.49	0.31
1.25-OHD [pg/mL]	48.51	14.95	49.51	16.71	0.45
VDBR [ng/mL]	30.28	14.40	29.63	8.75	0.62

**Table 3 ijms-24-03228-t003:** Calcium status before and after thyroidectomy.

Calcium Status	Before Surgery		After Surgery		*p*
	Median	IQR	Median	IQR	
Ca [mmol/L]	1.19	0.16	1.09	0.17	0.002

**Table 4 ijms-24-03228-t004:** Patient characteristics before thyroidectomy procedure.

Patients Characteristics *n* = 167	Median	IQR
Age [years]	48	24
Men [%]	29	-
Benign nodules [%]	79	-
Thyroid cancers [%]	21	-
Body mass index (BMI) [kg/m^2^], * (18.5–25)	27.76	7.79
TSH [uIU/mL], * (0.27–2.4)	1.29	0.78
Creatinine [mg/dL], * (0.5–0.9)	0.72	0.18
FT3 [pg/mL], * (2–4.4)	3.16	0.46
FT4 [ng/dL], * (0.93–1.7)	1.28	0.29
PTH [pg/mL], * (15–65)	19.7	22.18
Ca [mmol/L], * (1.0–1.3)	1.19	0.16
25-OHD [ng/mL], * (0–30)	6.13	2.51
1,25-(OH)_2_D [pg/mL] * (20–60)	48.69	15.17
VDBP [ng/mL]	30.12	14.13

* ()—normal range.

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
