# Peer review of "Vitamin D Status in Patients before Thyroidectomy"

_ijms, 2023, doi:10.3390/ijms24043228_

Round 1
Reviewer 1 Report
Paper “Vitamin D status in patients requiring thyroidectomy” brings important observations indicating on important role of vitamin D in recovery of the patients after thyroidectomy. The study, described in this paper, includes adequate methodology and good interpretation of results.
The determination of vitamin D levels prior to thyroidectomy may be useful for potential consideration of supplementation when vitamin D deficiency is revealed and help in better recovery of patients. However, further investigations are required to show a scale of vitamin D supplementation positive effect on patients’ recovery as study cohort included small number of patients who had normal vitamin D status.
Paper contains several minor uncertainties which require corrections:
· There is no explanation of BMI abbreviation in text – body mass index;
· In Table 1 – Age, Sex and the reason for thyroidectomy (nodule or cancer) are not biochemical parameters! In addition, percentage are indicated as median in this table. It is better to put these parameters in “Materials and Methods” section, in patients’ description;
· IQR is better to show in form of Q1-Q3 range, as it provides more information;
· The whole subsection about calcium concentration in “Results” section is confusing. First of all, there is a statement (line 118) that “Calcium concentration after surgery was significantly reduced (table 4) …”, however, median concentration shown in the table is significantly higher in comparison with data provided before surgery. May be, it is because in “Materials and Methods” section is no description about what method was used to determine calcium status in patients and therefore it should be provided. Indeed, if calcium concentration was detected in blood, authors should indicate whether this was ionic or total Ca detection.
Although, hypercalcemia is one of indicators of total calcium reduction in organism, there could be also other reasons, for example, increased absorption or tubular reabsorption resulting from a disorder such as primary hyperparathyroidism or hypervitaminosis D, or problems with kidneys [DOI: 10.1093/jn/120.suppl_11.1470]. Therefore, it is inappropriate to write about calcium concentration reduction, better to use another term, for example, “problem with calcium metabolism” or something similar;
· In “Discussion” section (line 184), again, authors write about hypocalcaemia, however, in Table 4 patients after surgery show hypercalcemia (high concentration of calcium, presumably, in peripheral blood);
· Text has minor punctuation and space problems.
Author Response
Response to reviews
I would like to thank all the reviewers for their comments and the time spent on the manuscript evaluation. Thank you for giving us the opportunity to revise and improve our work. The manuscript has been improved in accordance with the recommendations of most reviewers. However, we were unable to complete some of the suggestions. I have great hope that the improved version of our manuscript will meet the reviewers suggestions. All of the changes were written in red color.
1. We transfer table 1 to patient's characeristics section
2. We prefer to show the interquartile range as an IQR.
3. There was a mistake in table 4. We add proper information about calcium measurement in the materials and methods.
4. We changed "metabolism" to "reduction".
Reviewer 2 Report
It is a very interesting topic and easy for applicability.
Introduction needs evidence about expression levels of vitamin D in thyroid diseases and more references in last paragraph.
Line 39 need to be adjusted
Methods: Was there any history of vitamin or vitamin D supplementation regularly taken by any of the patients?
Results: Is there comorbidities that might influence vit D levels.
Please revise the values in table 4.
What is vitamin D postoperative? It is unclear with the current data that the association is related to the surgery as indicated by the theme of the article and title.
Author Response
Response to reviews
I would like to thank all the reviewers for their comments and the time spent on the manuscript evaluation. Thank you for giving us the opportunity to revise and improve our work. The manuscript has been improved in accordance with the recommendations of most reviewers. However, we were unable to complete some of the suggestions. I have great hope that the improved version of our manuscript will meet the reviewers suggestions. All of the changes were written in red color.
1. We improved the introduction section.
2. We added vitamin D supplementation information to the materials and method section.
3. We added the exclusion criteria to the patient's characteristics, and increased body weight was the only comorbidity associated with our study.
4. There was mistake in table 4.
5. We changed the title.
Round 2
Reviewer 2 Report
While I can see improvement in the article. There was no point to point response in the website available to screen.